# Nanoparticles Produced via Laser Ablation of Porous Silicon and Silicon Nanowires for Optical Bioimaging

**DOI:** 10.3390/s20174874

**Published:** 2020-08-28

**Authors:** Stanislav V. Zabotnov, Anastasiia V. Skobelkina, Ekaterina A. Sergeeva, Daria A. Kurakina, Aleksandr V. Khilov, Fedor V. Kashaev, Tatyana P. Kaminskaya, Denis E. Presnov, Pavel D. Agrba, Dmitrii V. Shuleiko, Pavel K. Kashkarov, Leonid A. Golovan, Mikhail Yu. Kirillin

**Affiliations:** 1Faculty of Physics, Lomonosov Moscow State University, 1/2 Leninskie Gory, 119991 Moscow, Russia; snastya.19996@mail.ru (A.V.S.); sea_nnov@yahoo.com (E.A.S.); kashaev.fedor@gmail.com (F.V.K.); ktp53@mail.ru (T.P.K.); denis-presnov@yandex.ru (D.E.P.); agrbapd@gmail.com (P.D.A.); dmitriy1815@gmail.com (D.V.S.); kashkarov@physics.msu.ru (P.K.K.); golovan@physics.msu.ru (L.A.G.); 2Institute of Applied Physics RAS, 46 Uljanov St., 603950 Nizhny Novgorod, Russia; vekfy@inbox.ru (D.A.K.); alhil@inbox.ru (A.V.K.); mkirillin@yandex.ru (M.Y.K.); 3Skobeltsyn Institute of Nuclear Physics, Lomonosov Moscow State University, 1/2 Leninskie Gory, 119991 Moscow, Russia; 4Quantum Technology Centre, Lomonosov Moscow State University, 1/35 Leninskie Gory, 119991 Moscow, Russia; 5Faculty of Radiophysics, N.I. Lobachevsky State University of Nizhny Novgorod, 23 Gagarin av., 603950 Nizhny Novgorod, Russia; 6Institute of Information Technology, Mathematics and Mechanics, N.I. Lobachevsky State University of Nizhny Novgorod, 23 Gagarin av., 603950 Nizhny Novgorod, Russia

**Keywords:** silicon nanoparticles, pulsed laser ablation in liquids, spectrophotometry, optical coherence tomography, fluorescence

## Abstract

Modern trends in optical bioimaging require novel nanoproducts combining high image contrast with efficient treatment capabilities. Silicon nanoparticles are a wide class of nanoobjects with tunable optical properties, which has potential as contrasting agents for fluorescence imaging and optical coherence tomography. In this paper we report on developing a novel technique for fabricating silicon nanoparticles by means of picosecond laser ablation of porous silicon films and silicon nanowire arrays in water and ethanol. Structural and optical properties of these particles were studied using scanning electron and atomic force microscopy, Raman scattering, spectrophotometry, fluorescence, and optical coherence tomography measurements. The essential features of the fabricated silicon nanoparticles are sizes smaller than 100 nm and crystalline phase presence. Effective fluorescence and light scattering of the laser-ablated silicon nanoparticles in the visible and near infrared ranges opens new prospects of their employment as contrasting agents in biophotonics, which was confirmed by pilot experiments on optical imaging.

## 1. Introduction

Modern trends in medicine imply development of novel treatment protocols based on theranostics, preferably, including non-invasive diagnostics modalities. Optical diagnostics techniques have high potential from this point of view and are currently actively introduced into clinical practice.

Nanosized agents have wide perspectives in developing theranostics approaches with assistance of optical imaging due to both existing instrumentation for controlling optical properties of fabricated nanoproducts and extensive capabilities of nanoparticles functionalization [1,2,3,4]. Due to small size, nanoparticles exhibit efficient penetration into biotissues, while conjugating them with specific antibodies allow for producing targeted nanoconstructs that demonstrate high contrast of accumulation in specific tissues, e.g., tumors. Endogenously activated silicon-based nanoconstructs may serve an efficient tool in tumor treatment [3]. High contrast in local optical properties ensured by nanoparticles accumulation allows to non-invasively control their biodistribution by means of optical imaging techniques. Light activation of the nanoconstructs accumulated in tumors provides an effective non-invasive treatment modality with minimal impact to surrounding normal tissues [2,4]. Many classes of nanoproducts, however, exhibit significant toxicity, which limits their application in biomedical studies. Silicon nanoproducts are a potential class of theranostics agents due to their biocompatibility, biodegradability, and specific optical properties [5,6], however, techniques for controlling physical and optical properties of silicon nanoparticles (SiNPs) require further improvement.

In recent years silicon nanoparticles (SiNPs) have been extensively used for theranostics applications. Nanoparticles-based platforms allow for precise targeted drug delivery [7,8] with the opportunity of both anticancer drugs and photothermal incorporation for the enhancement of therapeutic efficacy [9,10]. Photosensitizers based on SiNPs were shown to be efficient for photodynamic therapy of cancer both in vitro [11] and in vivo [12]. SiNPs are widely employed in fluorescence imaging either as contrasting agents [10,13,14] or sensors for quantitative measurements [15,16]. SiNPs-based contrast enhancement was demonstrated also in optical coherence tomography (OCT) [17,18] and magnetic resonance imaging [19]. Other SiNPs applications include tissue engineering [20], nonlinear-optical diagnostics [21], photo- [22] and ultrasound [23] hyperthermia.

Traditionally, porous silicon (PSi) structures are used in biophotonics applications [5,6]. A serious limitation for further advances in biomedicine is relatively large sizes of SiNPs provided by traditional fabrication techniques, such as mechanical milling [13,17,23,24] or ultrasonic grinding [11,25,26,27] of PSi matrices. Usually, the mean size of mechanically crushed powder particles of PSi exceeds 100 nm (although the nanocrystals themselves, which form these particles, may be smaller). This factor complicates SiNPs penetration in biotissues and further release from the organism. Alternative SiNPs produced by milling silicon nanowire (SiNW) arrays are characterized by similar sizes [28,29], which does not allow to overcome the limitation. Therefore, designing SiNPs with sizes smaller than 100 nm is one of actual challenges.

Pulsed laser ablation in liquids (PLAL) technique allows to overcome this limitation and provides small mean sizes (5–100 nm) and chemical purity for biomedical applications of SiNPs simultaneously [14,17,18,20,21,22,30]. However, to produce SiNPs with high concentration, this method requires either lasers with high pulse repetition rate and energy, or significant increase in exposure time. We suggest an alternative approach of using PSi [18] or SiNWs [31] targets instead of crystalline silicon ones. The pore presence decreases the mechanical strength and thermal conductivity of the targets. As a result, the ablation threshold is reduced sufficiently and SiNPs yield grows up to an order of magnitude.

Earlier we have demonstrated prospects of using SiNPs produced via PLAL of PSi as OCT [18] and fluorescence imaging [32] contrasting agents. The main aim of this work is a comprehensive study of structural, absorption, scattering, and fluorescence properties of SiNPs fabricated via PLAL of both target types: PSi films and SiNW arrays with different doping levels which defines the morphology of the targets [18,33,34].

## 2. Materials and Methods

The PSi targets for PLAL were fabricated by the anodic electrochemical etching technique [35]. We used one-side polished crystalline Si wafers (100) of two types for etching: boron-doped plates with the specific resistivities of 10–20 Ω·cm (low-doped) and 17–23 mΩ·cm (heavily-doped). Such approach provides fabrication of microporous (pore size smaller than 5 nm) and mesoporous (pore size of 5–70 nm) silicon layers, respectively [34]. Prior to etching, the plates were placed for 1 min in 47.5% hydrofluoric acid (HF) to remove natural oxide from the surface. An electrolyte consisting of a solution of hydrofluoric acid and ethanol (C_2_H_5_OH) in a ratio of 1:1 was used directly for etching. In the both cases the etching current density was 75 mA·cm^−2^, and the etching time was 30 min. After, the formed PSi layers were rinsed with distilled water to remove HF electrolyte remainders. The thicknesses of the fabricated layers were determined by means of inspecting the cross-sections of the targets cut through the center using an optical microscope Olympus BX41 to ensure the analysis of the area that underwent etching. The obtained thicknesses were 25 ± 3 μm and 80 ± 5 μm for the low-doped and heavily-doped samples, respectively.

The metal-assisted chemical etching (MACE) technique [33,36] was employed to fabricate Si-NW arrays that further served as targets for PLAL. The same two types of low-doped and heavily-doped Si wafers were etched with MACE. Preliminarily, all wafers were rinsed in the hydrofluoric acid for 1 min to remove surface oxide. Further, Ag nanoparticles were deposited on the silicon wafers by means of dipping the samples into a solution (1:1) of 0.02 M AgNO_3_ in water and 5 M HF for 30 s. At the next step, the silver-covered wafers were dipped into a solution (10:1) of 5 M HF and 30% hydrogen peroxide (H_2_O_2_) for 40 min. To remove the silver nanoparticles after etching all samples were placed into 65% nitric acid (HNO_3_) for 15 min and later rinsed with water. The fabricated SiNW arrays have thicknesses of 25 ± 4 μm and 10 ± 2 μm for low-doped and heavily-doped wafers, respectively.

Both PSi and SiNW layers (typical sample is shown in Figure 1a) were not detached from the silicon substrates after electrochemical etching and MACE, respectively. They were used as targets in PLAL directly on the substrates.

Irradiation of all PSi and SiNW targets was performed in a 15 mL cell filled with distilled water or ethanol using an EKSPLA PL 2143A picosecond Nd:YAG laser with a pulse repetition rate of 10 Hz (Figure 1b,c). The duration, wavelength, and energy of the pulses were 34 ps, 1064 nm, and 10 mJ, respectively. The exposure time varied in the range of 15–60 min to provide close to 0.5 mg/mL mass concentration of laser-ablated SiNPs in water and ethanol, which corresponds to a typical brown color of the suspensions (Figure 1d). The verification of this value was provided by gravimetric measurements after vaporization of buffer liquids from cuvettes containing the suspensions. Such concentrations provide a priori good biocompatibility: for example, porous silicon based SiNPs could penetrate into the cells without any cytotoxic effect up to the concentration of 100 mg/mL [37].

The laser beam was focused onto a layer of the Si-NWs by a lens with the focal distance of 40 mm at the normal incidence. To prevent degradation of the targets during PLAL, the cell was shifted perpendicular to the axis of the laser beam using two orthogonally oriented automated mechanical translation stages (Figure 1). To homogenize the produced SiNPs in suspensions during the ablation process, the liquid was mixed using an MM-1 magnetic stirrer.

After the PLAL the SiNPs were deposited on a pure silicon surface and were examined using a Carl Zeiss Supra 40 scanning electron microscope (SEM) and a ND-MDT SolverPRO scanning probe microscope in the atomic force microscopy (AFM) regime. Composition of the SiNPs was analyzed by measuring Raman spectra with a Horiba Jobin Yvon HR 800 spectrometer with excitation by Ar laser at the wavelength of 488 nm of the particles deposited on a glass substrate. High sensitivity in measuring small quantities of the SiNPs was ensured by using a 100× Olympus objective for this purpose.

The optical properties of the produced SiNP suspensions were determined in the wavelength range of 400–1000 nm based on spectrophotometric measurements of the collimated and diffuse transmission spectra, as well as diffuse reflectance spectra. The measurements were carried out for 5-mm thick SiNP suspension layers placed in a quartz cuvette using an Analytik Jena SPECORD 250 spectrophotometer equipped with an integrating sphere. Before all measurements, the suspensions were ultrasonicated to avoid SiNP agglomeration. The values of the absorption *μ*_a_ and scattering *μ*_s_ coefficients as well as the anisotropy factor *g* were reconstructed from the measured collimated transmittance $T_{C}^{meas}$, diffuse transmittance $T_{D}^{meas}$, and reflectance $R_{D}^{meas}$ using a lookup table based on Monte Carlo simulations. Monte Carlo method, implying modelling of a random photon trajectory in turbid media, allows simulating light transport in a sample during spectrophotometric measurements. The lookup table accumulates the results of Monte Carlo numerical simulations for various combinations of the sample properties ($\mu_{a}^{i}$, $\mu_{a}^{i}$, $g^{i}$) performed with predefined steps (*i* is the number of the predefined optical properties vector) in each value and provides a discrete function that returns corresponding coefficients ($T_{C}^{i}$, $T_{D}^{i}$, $R_{D}^{i}$) for each vector of the sample optical properties values. The inverse function with argument of (*T_C_*, *T_D_*, *R_D_*) vector based on minimization of the functional
(1)$$F\left(T_{C}^{i},\;T_{D}^{i},\;R_{D}^{i}\right)≔\frac{{\left(T_{C}^{i}-T_{C}^{meas}\right)}^{2}}{{\left(T_{C}^{meas}\right)}^{2}}+\frac{{\left(T_{D}^{i}-T_{D}^{meas}\right)}^{2}}{{\left(T_{D}^{meas}\right)}^{2}}+\frac{{\left(R_{D}^{i}-R_{D}^{meas}\right)}^{2}}{{\left(R_{D}^{meas}\right)}^{2}}\;\to \underset{\left[T_{C}^{i},\;T_{D}^{i},\;R_{D}^{i}\right]}{\min}$$
that returns (*μ*_a_, *μ*_s_, *g*) vector is employed for optical properties reconstruction.

The experiments on imaging of biological tissue phantoms using OCT in the presence of SiNPs, which serve as contrast agents, were performed using an OCT-1300E system (central probing wavelength of 1300 nm, IAP RAS, Biomedtech, Russia). Droplets of SiNP suspensions were administered to the surface of a phantom represented by a 0.2% agar gel.

Measurements of fluorescence spectra of PSi layers and SiNPs suspensions in the range of 500–1100 nm were carried out using a Princeton Instruments SpectraPro 2500i spectrograph upon excitation by the second harmonic (532 nm, 10 μJ) from the Nd:YAG laser employed for ablation. The suspension was placed in a quartz cuvette of 3 mm in thickness, which exhibits no fluorescence under employed excitation. All fluorescence spectra were corrected for the wavelength-dependent detector sensitivity with the help of a temperature lamp TRSh 2850–3000. In this study, we focused on visible fluorescence band, since it is directly determined by SiNPs and is much more effective than the infrared one, which is caused by indirect transitions in crystalline silicon or SiNPs. Fluorescence images of the SiNP suspensions were obtained using custom-made dual-wavelength fluorescence imaging device (IAP RAS, Russia) [38]. The device is equipped with two LED sources at the wavelengths of 405 ± 10 nm (blue) and 660 ± 10 nm (red) synchronized with the CCD camera with the Semrock 772/140 bandpass detection filter (700–850 nm). Additionally, to observe visible fluorescence by naked eye a continuous He-Cd laser GKKL-30 UM(I) with wavelength of 325 nm and output power of 30 mW was used as an excitation source.

## 3. Results and Discussion

### 3.1. Structural Properties

Employment of different etching techniques and silicon wafers with different doping levels allowed to fabricate silicon nanostructured targets for PLAL with different morphology. Microporous silicon as first type of these targets contains a lot of random pores including those of nanometer size as shown in Figure 2a. On the contrary, mesoporous silicon possesses the distinguished pore orientation direction (Figure 2b) that corresponds to [100] crystallographic axis [34]. These pores can be treated as canals with a sufficiently large diameter of the order of tens of nanometers and a high aspect ratio.

SiNW arrays of both types consist of the wires with diameters in the range of 100–200 nm oriented perpendicular to the substrate surface (Figure 2c,d). However, in case of heavily-doped SiNWs their top endings tend to agglomerate (Figure 2d) in contrast to the separated low-doped wires (Figure 2c). For explaining this behavior, we assume the existence of a relatively strong electrostatic forces in the heavily-doped structures.

After PLAL, SEM study of the fabricated Si-NPs revealed their substantially spherical shape in all cases (see, for example, Figure 3a). The reason of this phenomenon is in the existence of ablation products in a melting state after ablation before agglomeration into nanoparticles [39].

Raman analysis revealed long-range crystalline order in the SiNPs. A typical Raman spectrum (Figure 3b) demonstrates the narrow line near 521 cm^−1^ corresponding to the crystalline silicon phase and absence of both an amorphous fraction which is usually manifested by the wide line at 480 cm^−1^ [40] and other polymorphous Si fractions [41]. Most likely, the SiNPs are silicon nanocrystals or have polycrystalline structure.

Quantitative study of SiNP size distributions was performed by means of AFM data analysis. The results are presented in Table 1 and Figure 4, where size is the maximal cross section diameter of SiNP. All SiNPs have the mean size smaller than 70 nm with polydisperse distributions which are characterized by standard deviation values in the range of 9–24 nm. Consequently, the obtained SiNPs are noticeably smaller than those fabricated by mechanical milling or ultrasonic grinding of PSi and SiNW matrices [11,13,17,23,24,25,26,27,28,29].

The SiNPs produced by PLAL of microporous silicon (Figure 4a,b) demonstrate larger mean sizes in comparison to the case of ablation of mesoporous silicon (Figure 4c,d and Table 1). This difference can be explained by larger pore size reaching up to 70 nm and co-directed canal-like structures in mesoporous silicon (Figure 1b) in contrast to the more solid isotropic microporous silicon structure (Figure 1a). The latter structures have undergone less destruction under laser pulse exposure and laser ablation products agglomerate to larger SiNPs.

Additionally, the particles formed by ablation in water from microporous silicon demonstrate a pronounced bimodal size distribution. This phenomenon was earlier registered at pulsed laser ablation of crystalline silicon in water [18] and can be explained by the theory developed by Shih et al. [42]. The appearance of smaller particles is due to the fast nucleation of atoms evaporated from the target surface in the area of interaction with water molecules under continuous in time tendency towards thermodynamic equilibrium (cooling of ablation products) [39]. At the same time the larger particles originate from the presence of Richtmyer-Meshkov thermodynamic instability [43] resulting from exposure by a short laser pulse [44] and leading to a shock ejection of the melt in the form of nanojets from the overheated target, at the end of which droplets appear to agglomerate when cooled to sufficiently large nanoparticles. On the contrary, in the case of laser ablation of mesoporous silicon in water the corresponding histogram does not reveal two distinguished peaks (Figure 4c). According to the made above assumption about more effective destruction of such PSi layers in the course of PLAL, the overheat state, which is responsible for formation of larger SiNPs, is less probable.

The SiNPs produced by PLAL of SiNW targets do not manifest significant dependence of the size distributions on the substrate doping level (Figure 4e–h). However, the mean sizes are larger in case of ablation in water in comparison to those for ablation in ethanol (Table 1). Actually, the low-doped and heavily doped SiNW targets have similar morphology (Figure 2c,d), however, the boiling temperature of water and ethanol differ substantially amounting at the normal atmospheric pressure 100 and 78 °C for water and ethanol, respectively. During PLAL the liquids are boiled near target surfaces, that induces formation of many cavitation bubbles [39]. Due to the smaller boiling temperature, in ethanol these processes are more effective, which results in more distant removal of PLAL products (silicon atoms and droplets) away from target in comparison to ablation in water with the same energy fluence of laser pulses. As a result, the concentration of ablation products in the buffer liquid is smaller, which leads to smaller agglomeration efficiency and, hence, smaller particle size.

Analysis of the mean sizes of the SiNPs formed by means of picosecond laser ablation of the PSi and SiNW based targets in water and ethanol (Table 1) allows concluding that they are promising for biomedical applications. The relatively small mean size of SiNPS ensures their effective penetration in living organisms, moreover, it can be customized in the range of 14–65 nm in the course of fabrication by an appropriate choice of the targets and buffer liquids. Prevalence of the crystalline phase in the fabricated particles ensures their high refractive index (≈3.6) [45] and fluorescence emission [32] in the so-called biotissue diagnostic transparency window (700–1300 nm), which indicates their high potential as scattering and/or fluorescent contrast agents in optical bioimaging.

### 3.2. Spectrophotometry and OCT

Figure 5 shows reconstructed spectra of absorption and scattering coefficients of the fabricated SiNP suspensions in water and ethanol. The absorption coefficient *μ*_a_ for all SiNP suspensions monotonically decreases in the visible range (400–750 nm) from values of the order of 0.1 mm^−1^ to values close to zero. This behavior is in good agreement with the spectral dependence of silicon absorption [45,46]. The insignificant absorption peak near 970 nm for the cases of ablation in water (Figure 5a,c,e,g) corresponds to absorption peak of water, which serves as a host liquid of the suspensions. At the same time, for SiNPs fabricated and subsequently remaining in ethanol, this peak is not observed (Figure 5b,d,f,h). Weak absorption of the studied SiNP suspensions at wavelengths above 700 nm confirms their prospects in applications in biomedical imaging techniques employing light scattering in the near infrared range or fluorescence excitation in the blue-green range with effective absorption of incident photons and consequent fluorescent emission at longer wavelengths.

Spectral dependencies of the scattering coefficient *μ*_s_ for all SiNP suspensions demonstrate the values in the range of 0.02–0.5 mm^−1^ for the spectral region of 400–1000 nm (Figure 5), which seems to be promising for effective contrasting in OCT technique. In the shorter wavelength range of 400–550 nm these spectra demonstrate weak dependence on wavelength with some exclusion for cases of mesoporous silicon in ethanol and heavily-doped SiNWs in water (Figure 5d,g). At longer wavelengths (550–1000 nm) a distinguished monotonous decrease is observed in all cases. The increasing wavelength dependence of the scattering coefficient with the wavelength value increase can be explained by the increasing wavelength dependence when transiting from Mie scattering (scattering cross-section *σ* ∝ *λ*^−2^) to Rayleigh scattering (*σ*_s_ ∝ *λ*^−4^) [47]. At shorter wavelength Mie scattering plays a key role because the ratio of the SiNP size *d* to the wavelength *λ* is larger than that at longer wavelengths, while Rayleigh scattering regime starts to dominate in the red and near infrared spectral regions.

The transition region between Mie and Rayleigh scattering can be estimated around 550 nm for the considered suspensions. It allows defining typical SiNP diameter *d* as 37 nm as the upper limit in the Rayleigh scattering condition *d*/*λ* ≲ 1/15 [47]. This size is close to the calculated mean sizes for SiNPs (Table 1), that agrees with our assumption. However, it is worth mentioning that due to polydisperse nature of the fabricated SiNP suspensions (Figure 4), which may feature one or two peaks in the size distributions (see Figure 4), the spectra of the overall suspensions exhibit quite complex behavior. Moreover, even in the near infrared range (800–1000 nm) the scattering regime cannot be considered as Rayleigh only, which is confirmed by the weaker wavelength dependence of *μ*_s_ as compared to ∝*λ*^−4^. The absolute value of the wavelength power dependence factor does not exceed 3.3 in accordance with our estimations (see, for example, approximation in Figure 5c).

It is worth mentioning that the concentrations of the silicon nanoparticles in biotissues may potentially significantly exceed those achieved in the suspensions (≈0.5 mg/mL), when after administration the liquid is removed, while the nanoparticles are accumulated in the target area. This may result in significantly higher partial scattering coefficient as compared to that measured in suspension, ensuring high scattering contrast capable of detection with OCT imaging technique. The perspectives of using SiNPs, fabricated via laser ablation, in OCT imaging were demonstrated in our model experiments with the nanoparticles administered to an agar phantom. Figure 6a shows a typical OCT image of the initial agar phantom, characterized by a small OCT contrast value. The upper bright line corresponds to the surface of the fiber probe, while the lower line corresponds to the air–agar phantom interface. The area below the second line corresponds to the signal from the phantom and is characterized by a low OCT signal level, which amounted 7.7–8.3 dB for all used samples without SiNPs.

Figure 6b,c shows OCT images of the phantom after topical administration of a droplet of a SiNP suspension produced by ablation of PSi in water and ethanol, respectively. The OCT signal from the suspension droplet on agar phantom surface amounts 14.6 and 16.6 dB for mesoporous silicon ablation in water and ethanol, correspondingly, confirming their potential in detection by OCT.

### 3.3. Fluorescence

Our recent pilot studies revealed capability of SiNPs produced via laser ablation of PSi in ethanol and liquid nitrogen to emit fluorescence in the spectral range of 600–900 nm [32] which is suitable for optical bioimaging. Thus, a more comprehensive search for fluorescent SiNPs formed by means of PLAL both PSi and SiNW targets is promising.

Figure 7 presents fluorescence spectra of the SiNPs formed by laser ablation of micro- and mesoporous silicon in water and comparative spectra of the initial targets. The initial microporous silicon layer demonstrates bright visible (550–800 nm) fluorescence (Figure 7a). It is the so-called S-band fluorescence caused by quantum confinement in Si nanocrystals [48,49,50]. The nanocrystal size determines the fluorescence peak wavelength, which grows with the SiNP size increase.

They also demonstrate visible fluorescence; however, its spectrum is shifted to red zone with maximum at 750 nm (Figure 7a). Similar broad and red-shifted fluorescence spectra were previously found for SiNPs formed by laser ablation of microporous silicon in ethanol [32]. Such features could be caused by the following reasons:The polydisperse size distribution of SiNPs (Figure 4a) provides presence of excitons responsible for fluorescence with different binding energies in Si nanocrystals [48].Laser ablation of microporous silicon layers results in formation of the relatively large SiNPs with sizes more than 10 nm (Figure 4a) in contrast to the initial targets nanostructured at smaller scale [34]. Therefore, the formed SiNPs are characterized by less fluorescence photon energy (larger emission wavelength).

In contrast to microporous silicon, laser ablation of mesoporous silicon causes blue shift of fluorescence spectrum relative to that for an initial layer (Figure 7b). Wide fluorescence peak is situated near 650 nm. These features for mesoporous silicon-based nanostructures agree with the proposed above explanation of regularities for fluorescence spectra. These SiNPs have smaller sizes than those ablated from microporous silicon (Figure 4a,c and Table 1) and crystallites forming mesoporous matrix (Figure 2b).

It is necessary to mention that the initial mesoporous layers exhibit two order of magnitude lower fluorescence efficiency in comparison with the microporous silicon ones owing to less exciton energy and higher free-carrier concentration (cf. Figure 7a,b) [48]. Although fluorescence of the thin SiNP layer deposited on the substrate was found to be more effective than that for the thicker initial mesoporous silicon layer (Figure 7b), it is comparable with the fluorescence intensity of SiNPs laser-ablated from microporous silicon. However, due to its red fluorescence shift, the latter one is preferable for diagnostics of biological tissues owing to their better transparency in the 700–800 nm wavelength range in comparison to shorter wavelengths.

In contrast to the case of PSi, the shape and the peak position in the fluorescence spectra of SiNPs produced by laser ablation of SiNW arrays in water and ethanol demonstrate smaller difference with those of the initial SiNW arrays (see, for example, Figure 8a,b). Blue shift of fluorescence peak after ablation does not exceed 30 nm in all cases and, most likely, is caused by transformation of SiNWs into SiNPs with reducing typical sizes of the structures as can be seen from comparison of respective SEM images (Figure 2c,d) and size distribution histograms (Figure 4e–h).

Much more effective fluorescence of heavily-doped SiNWs in comparison with low-doped SiNWs results in much more effective fluorescence of the SiNPs formed by its ablation (cf. Figure 8a,b). The fluorescence of the SiNPs in suspension can be easily seen with the naked eye (reddish trace in the cuvette in Figure 9a) for the SiNPs fabricated from heavily-doped SiNWs. However, this pronounced effect is absent in SiNPs formed by laser ablation of low-doped SiNWs, indicating that the fluorescence yield is weaker in the latter case. Fluorescence imaging of plastic Eppendorf cuvettes filled by the studied suspensions confirmed this statement (Figure 9b). As one can see, only suspensions of SiNPs formed by ablation of heavily doped SiNWs exhibit detectable fluorescence level (cuvettes 2 and 4). The difference, most likely, is due to presence of numerous silicon nanocrystals of about several nanometers in size, which cover the heavily-doped SiNW surface just before ablation and stay preserved after ablation, that are responsible for effective visible fluorescence [28,51]. In the case of low-doped SiNWs the sample surface is smooth, without such nanocrystals. In its turn, the difference of the images for the SiNPs formed by ablation of SiNWs in water and ethanol (cuvettes 2 and 4) is presumably caused by their partial oxidation in water and ethanol suspensions.

It is necessary to mention the control of the SiNPs stability should be provided prior to using them in bioimaging. Although, before all measurements the suspensions were sonicated, the SiNPs tend to agglomerate with time and may be characterized by just moderate stability. This tendency is clearly seen in Figure 9b, where the maximal fluorescence signal is observed from the cuvette bottom. Our measurements of zeta potentials with a dynamic light scattering analyzer Malvern Zetasizer Nano ZS revealed that the SiNPs ablated in ethanol possess lower stability in comparison to those ablated in water: absolute values of the zeta potential for these cases are of order 10 and 30 mV, respectively. From this point of view, the suspensions of SiNPs ablated in water are more stable and are preferable for imaging applications.

Thus, PLAL of PSi and SiNW layers results in formation of SiNPs with fluorescence in the red spectral region. Whereas PLAL of PSi results in significant modification of the fluorescence spectra as compared to those of initial targets, the spectra of the SiNPs produced by PLAL of SiNWs differ insignificantly from the targets. These variations could be associated with effects of the SiNP size distributions and doping level of ablation targets on spectra and efficiency of SiNP fluorescence. It looks promising to employ PSi and SiNW targets with more efficient fluorescence (microporous silicon and heavily-doped SiNWs) to get brighter fluorescence of the laser-ablated SiNPs in contrasting images of biotissues and their phantoms.

## 4. Conclusions

We demonstrated capabilities of the technique for SiNP formation by picosecond laser ablation of PSi layers and SiNW arrays in water and ethanol. The fabricated particles are characterized by mean sizes in the range of 14–65 nm, which are determined both by the buffer liquid and the specific resistivity (doping level) of the initial crystalline silicon substrates. These sizes are substantially smaller than those for SiNPs obtained with the traditional fabrication techniques: mechanical milling and ultrasonic grinding of PSi and SiNW matrices. At the same time, the reported SiNPs, most likely, possess crystalline or polycrystalline structure confirmed by the analysis of Raman spectra.

The relatively small sizes together with crystalline like structure open new prospects for the employment of the laser-ablated nanocrystals in optical biomedical imaging techniques based on scattering and fluorescence detection principles. The fabricated SiNPs demonstrate low absorbance and effective light scattering in the spectral region of 700–1000 nm, which is within the diagnostic transparency window for biological tissues. Experiment with an agar gel-based phantom demonstrated high contrasting properties of the SiNPs formed via PLAL of PSi in OCT imaging.

All studied SiNPs demonstrate fluorescence with the emission peak situated in the range of 650–750 nm depending on the SiNP sizes, doping level, and etching technology of the used targets for ablation. These features fit well for fluorescence imaging of biological tissues and their phantoms. SiNPs produced via PLAL of microporous silicon formed by electrochemical etching of low-doped silicon and SiNWs fabricated by MACE of heavily-doped substrates demonstrate the highest fluorescence efficiency and seem to be preferable for optical imaging applications.

## Figures and Tables

**Figure 1 sensors-20-04874-f001:**
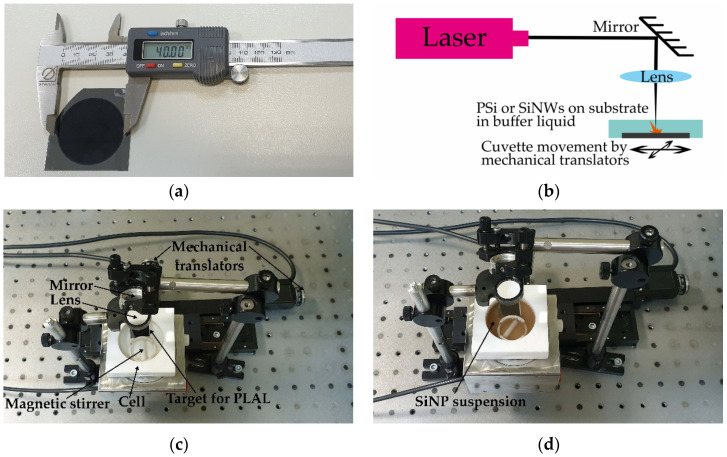
(**a**) Mesoporous silicon layer (17–23 mΩ·cm) on a crystalline silicon substrate; (**b**) Scheme of pulsed laser ablation in liquids (PLAL); Mesoporous silicon target in the cell with water (**c**) before and (**d**) after 30 min ablation.

**Figure 2 sensors-20-04874-f002:**
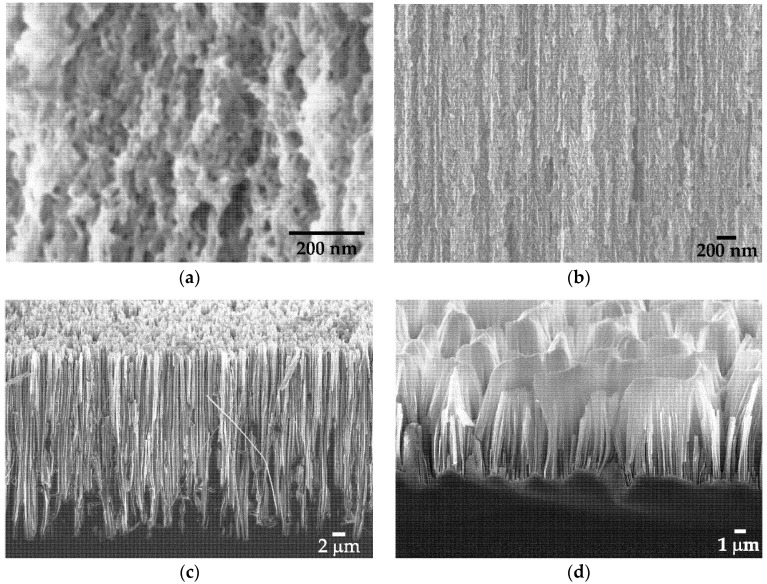
Side view scanning electron microscope (SEM) images of (**a**) microporous silicon; (**b**) mesoporous silicon; (**c**) low-doped and (**d**) heavily-doped silicon nanowire (SiNW) arrays.

**Figure 3 sensors-20-04874-f003:**
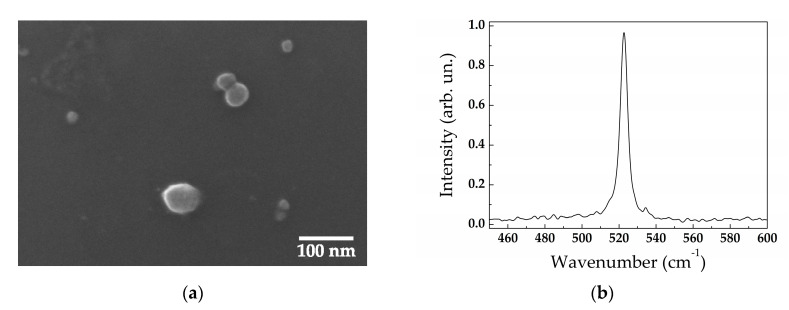
(**a**) Typical SEM image of silicon nanoparticles (SiNPs) produced by laser ablation of low-doped SiNW arrays in water and (**b**) their Raman spectrum.

**Figure 4 sensors-20-04874-f004:**
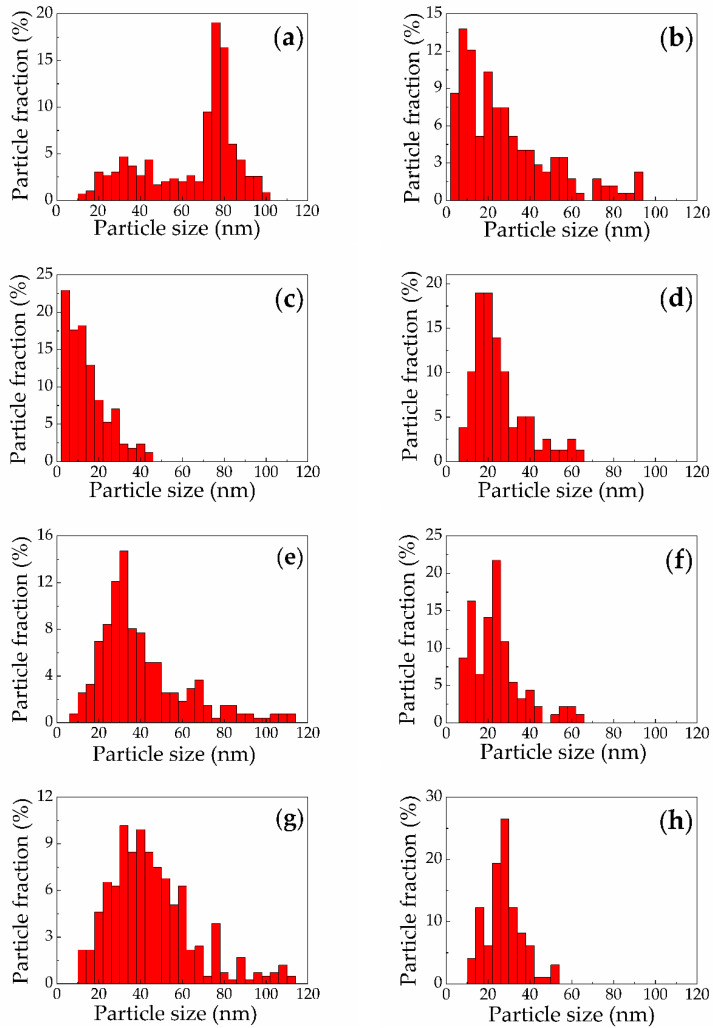
Size distributions for SiNPs fabricated via laser ablation of (**a**) microporous silicon in water; (**b**) microporous silicon in ethanol; (**c**) mesoporous silicon in water; (**d**) mesoporous silicon in ethanol; (**e**) low-doped SiNWs in water; (**f**) low-doped SiNWs in ethanol; (**g**) heavily-doped SiNWs in water; (**h**) heavily-doped SiNWs in ethanol.

**Figure 5 sensors-20-04874-f005:**
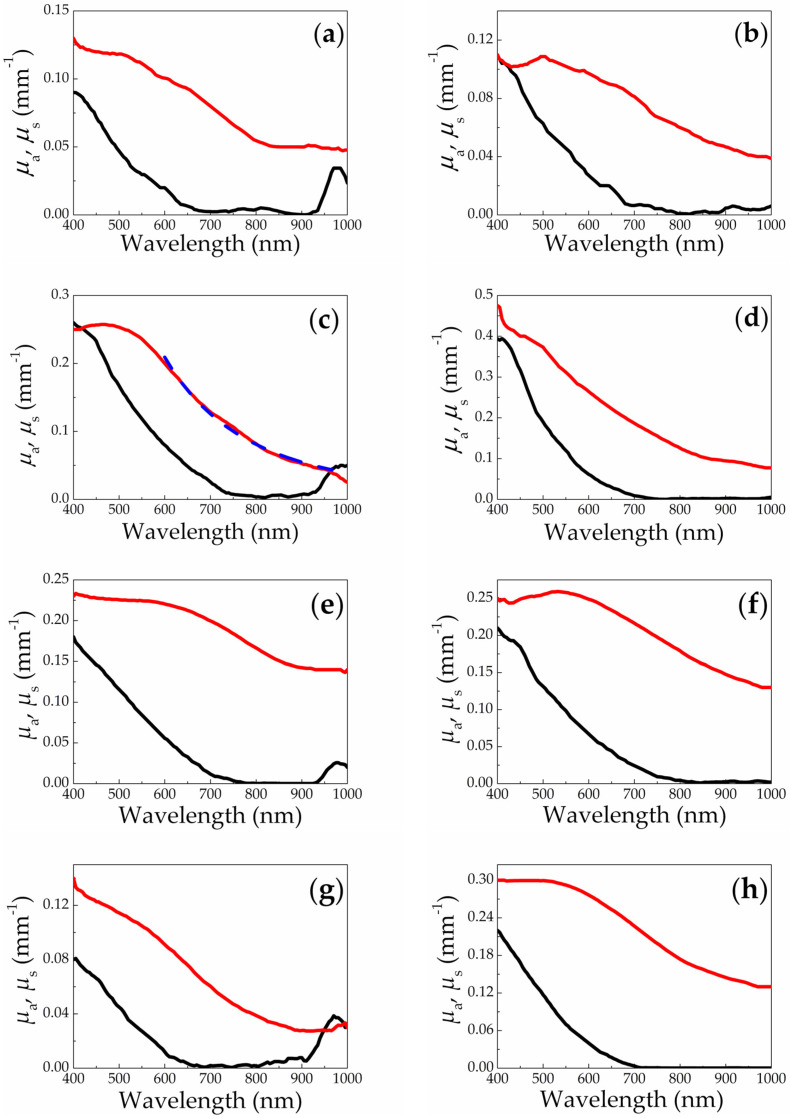
Spectra of absorption (*μ*_a_—black line) and scattering (*μ*_s_—red line) coefficients for SiNPs fabricated via laser ablation of (**a**) microporous silicon in water; (**b**) microporous silicon in ethanol; (**c**) mesoporous silicon in water, dashed blue line is an approximation by power dependence *λ*^−3.3^; (**d**) mesoporous silicon in ethanol; (**e**) low-doped SiNWs in water; (**f**) low-doped SiNWs in ethanol; (**g**) heavily-doped SiNWs in water; (**h**) heavily-doped SiNWs in ethanol.

**Figure 6 sensors-20-04874-f006:**
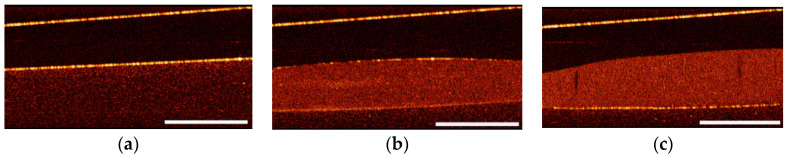
In-depth optical coherence tomography (OCT) images of agar gel phantoms (**a**) without nanoparticles; with the topically administered droplet of SiNP suspension formed by laser ablation of mesoporous silicon in (**b**) water, and (**c**) ethanol. The scale bar length equals 1 mm.

**Figure 7 sensors-20-04874-f007:**
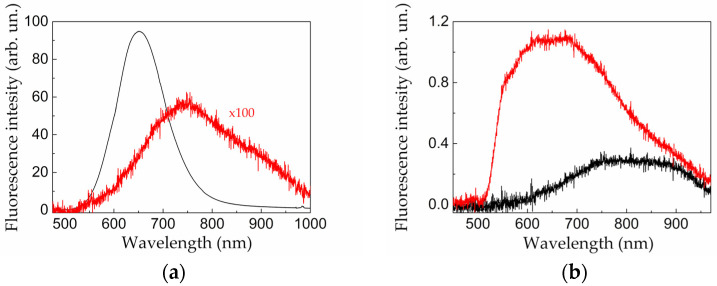
Fluorescence spectra of suspensions of SiNPs formed by laser ablation technique in water from (**a**) microporous and (**b**) mesoporous silicon layers (red lines) and spectra of the initial targets (black lines). Emission was excited by laser pulses at 532 nm. All graphs are shown at the same fluorescence intensity scale.

**Figure 8 sensors-20-04874-f008:**
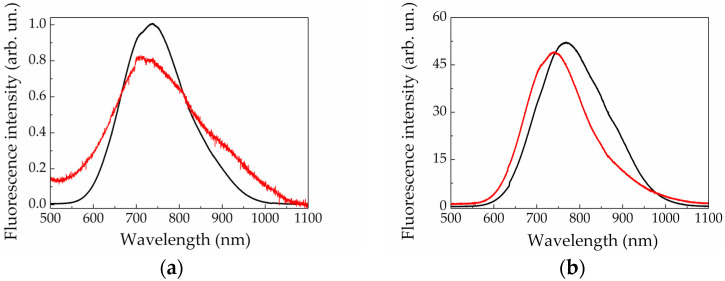
Fluorescence spectra of the initial SiNW arrays (black lines) of (**a**) low- (**b**) and heavily-doped Si fabricated by metal-assisted chemical etching (MACE) and corresponding SiNP suspensions (red lines) formed by their laser ablation in ethanol. Emission was excited by laser pulses at 532 nm. Both graphs are normalized to fluorescence maximum of the low-doped SiNW array.

**Figure 9 sensors-20-04874-f009:**
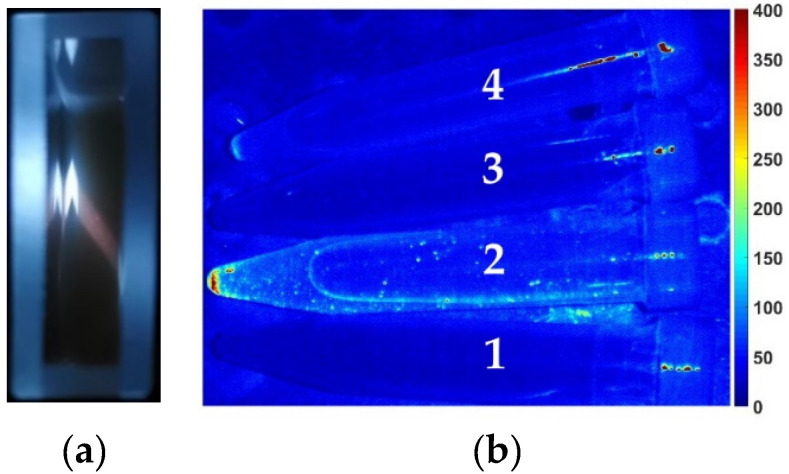
(**a**) Top view of the cuvette with suspension of the SiNPs formed by ablation of heavily-doped SiNWs in ethanol, excitation at 325 nm; (**b**) Fluorescence image (excitation at 405 nm, detection at 700–850 nm) of cuvettes with SiNPs suspensions fabricated by laser ablation of low-doped SiNWs in water (1), heavily-doped SiNWs in water (2), low-doped SiNWs in ethanol (3), and heavily-doped SiNWs in ethanol (4).

**Table 1 sensors-20-04874-t001:** Mean size and size standard deviation of SiNPs produced via PLAL using different targets and buffer liquids.

Target Type	Buffer Liquid	SiNP Mean Size (nm)	Size Standard Deviation (nm)
Microporous silicon	Water	65	22
Microporous silicon	Ethanol	28	22
Mesoporous silicon	Water	14	10
Mesoporous silicon	Ethanol	25	13
Low-doped SiNWs	Water	42	24
Low-doped SiNWs	Ethanol	24	9
Heavily-doped SiNWs	Water	45	20
Heavily-doped SiNWs	Ethanol	28	9

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
