# Peer review of "Nanoparticles Produced via Laser Ablation of Porous Silicon and Silicon Nanowires for Optical Bioimaging"

_sensors, 2020, doi:10.3390/s20174874_

Round 1

Reviewer 1 Report

The paper is very interesting and well prepared. In my opinion it is ready for printing.

Author Response

The paper is very interesting and well prepared. In my opinion it is ready for printing.

Response: We thank the reviewer for high evaluation of the importance and readiness of our manuscript.

Reviewer 2 Report

In the manuscript, fabrication of Si nanoparticles from various Si templates by laser ablation method is described. The nanoparticles are characterized by very small sizes of the order of tens of nanometers which makes them suitable for optical bioimaging. The manuscript is clearly written. A few minor points should be addressed before publication:

  • Page 10 on the top. It should be ‘Figures 6b and 6c’, not ‘Figures 4b and 4c’
  • Can we compare the intensities of the fluorescence spectra in Figs 7a, 7b, 8a and 8b? E.g., is the fluorescence intensity 50 times larger in the case of spectrum presented in Figure 8a compared to the spectrum from Figure 8b? If the tick labels on the Y-scale are not significant in these Figures, please, delete them.     
  • In Figure 9b, there is no emission visible from the nanoparticles in cuvette 1. On the other hand, the corresponding spectrum is characterized by a quite high emission intensity, Figure 8a red line. Could you explain it?
  • Is the fluorescence intensity drop at wavelengths larger than 900nm in Figs 7 and 8 a real effect? Or due to sensitivity of the detector or any other element of the optical setup?

Author Response

Response to Reviewer 2 Comments

We thank the reviewer for high evaluation of the importance of our manuscript.

We have improved the manuscript text in accordance with the comments.

Point 1: Page 10 on the top. It should be ‘Figures 6b and 6c’, not ‘Figures 4b and 4c’

Response 1: It is a technical misprint. We corrected the figures numbers (page 10, line 312).

Point 2: Can we compare the intensities of the fluorescence spectra in Figs 7a, 7b, 8a and 8b? E.g., is the fluorescence intensity 50 times larger in the case of spectrum presented in Figure 8a compared to the spectrum from Figure 8b? If the tick labels on the Y-scale are not significant in these Figures, please, delete them.

Response 2: Indeed, we can compare intensities of the fluorescence spectra in Figs. 7a and 7b and Figs. 8a and 8b, respectively. We do believe that the tick labels are significant in these figures, that is why we refused earlier proposed normalization of the spectra and corrected the Y-scale values in Figs. 7a and 7b and add ‘x100’ mark in Fig. 7a to make the quantitative comparison of the fluorescence signals possible. We specially emphasize that the fluorescence spectra for the ablated SiNPs were measured in suspensions in quartz cuvettes (page 4, lines 164–165, ‘The suspension was placed in a quartz cuvette of 3 mm in thickness, which exhibit no fluorescence under employed excitation.’). In fact, fluorescence intensity for the suspension of SiNPs formed by laser ablation of heavily-doped SiNWs is 50 times higher than that for the suspension of SiNPs formed by laser ablation of low-doped SiNWs.

Point 3: In Figure 9b, there is no emission visible from the nanoparticles in cuvette 1. On the other hand, the corresponding spectrum is characterized by a quite high emission intensity, Figure 8a red line. Could you explain it?

Response 3: Cuvette 1 is filled with products of laser ablation of low-doped SiNWs in water. Please note that Fig. 8 presents fluorescence spectra of SiNPs formed by means of laser ablation the same SiNWs in another medium, namely ethanol. Nevertheless, the results presented in Fig. 9 also demonstrate much higher efficiency of the fluorescence in the SiNPs formed by laser ablation of heavily doped Si. This fact was mentioned in the text on page 12, lines 365–367: ‘Much more effective fluorescence of heavily-doped SiNWs in comparison with low-doped SiNWs results in much more effective fluorescence of the SiNPs formed by its ablation (cf. Figure 8a and Figure 8b).’

Point 4: Is the fluorescence intensity drop at wavelengths larger than 900nm in Figs 7 and 8 a real effect? Or due to sensitivity of the detector or any other element of the optical setup?

Response 4: The obtained fluorescence spectra were corrected to take into account spectral function of the detector and optical elements of the experimental setup by means of measuring calibration function with the help of etalon light source (temperature lamp at 2850 К). We mentioned it in part 2. Materials and Methods as ‘All fluorescence spectra were corrected for wavelength-dependent detector sensitivity with the help of temperature lamp TRSh 2850-3000.’ (page 4, lines 165–166). Thus, the fluorescence intensity drop at wavelengths larger than 900nm in Figs 7 and 8 is a real effect. Our preliminary results indicate that the PSi and SiNWs exhibit infrared fluorescence caused by indirect transitions in silicon. Nevertheless, it would be more instructive to detect visible fluorescence, which is determined directly by the SiNPs and is of much higher magnitude. We add a brief discussion in part 2. Materials and Methods (page 4, lines 166–168): ‘In this study, we focused on visible fluorescence band, since it is determined by SiNPs and much more effective than infrared one, which is caused by indirect transitions in crystalline silicon or SiNPs.’

Additionally, we found and corrected some misprints in the text and improved English.

Reviewer 3 Report

The Authors have submitted a manuscript regarding the synthesis of silica NPs by laser ablation for imaging purposes.

The manuscript is interesting and the claims seem supported. The presentation is professional. Overall, I suggest the acceptance of this ms after the following minors are addressed:

-Line 49-51: "Light activation..tissues". Also endogenously activated NPs "provides an effective non-invasive treatment modality". Authors should rephrase the sentence and add significant references (for example on biodegradable silica NPs as, doi: 10.3390/cancers12051063).

-Table 1: errors are required

-Fig.9: Authors should add exps on colloidal stability. From Fig.9 seems NPs are precipitated and not omogeneously dispersed. 

Author Response

Response to Reviewer 3 Comments

We thank the reviewer for high evaluation of the importance of our manuscript.

We have improved the manuscript text in accordance with the comments.

Point 1: Line 49–51: "Light activation..tissues". Also endogenously activated NPs "provides an effective non-invasive treatment modality". Authors should rephrase the sentence and add significant references (for example on biodegradable silica NPs as, doi: 10.3390/cancers12051063).

Response 1: We agree that this part of the text requires extension regarding endogenous activation and adding references. We added 4 new references [1–4] including the suggested one [3]. The changes are presented on page 2, lines 44–54 as ‘Nanosized agents have wide perspectives in developing theranostics approaches with assistance of optical imaging due to both existing instrumentation for controlling optical properties of fabricated nanoproducts and extensive capabilities of nanoparticles functionalization [1–4]. … Endogenously activated silicon-based nanoconstructs may serve an efficient tool in tumor treatment [3]. … Light activation of the constructs targetly accumulated in tumors provides an effective non-invasive treatment modality with minimal impact to surrounding normal tissues [2,4].’

Point 2: Table 1: errors are required.

Response 2: We calculated and added in Table 1 values of the standard deviation for the obtained size distributions for laser-ablated silicon nanoparticles. The corresponding text was changed on page 6, lines 203–204 as ‘All SiNPs have the mean size smaller than 70 nm with polydisperse distributions which are characterized by standard deviation values in the range of 9 – 24 nm.’

Point 3: Authors should add exps on colloidal stability. From Fig.9 seems NPs are precipitated and not homogeneously dispersed.

Response 3: Indeed, the problem of colloidal stability is important in such studies. We added discussion about it on page 13, lines 385–393:

‘It is necessary to mention, the control of the SiNPs stability should be provided prior to using them in bioimaging. Although, before all measurements the suspensions were sonicated, the SiNPs tend to agglomeration with time and may be characterized by just moderate stability. This tendency is clearly seen in Figure 9b, where the maximal fluorescence signal is observed from the cuvette bottom. Our measurements of zeta potentials with a dynamic light scattering analyzer Malvern Zetasizer Nano ZS revealed that the SiNPs ablated in ethanol possess lower stability in comparison to those ablated in water: absolute values of the zeta potential for these cases are of order 10 mV and 30 mV, respectively. From this point of view, the suspensions of SiNPs ablated in water are more stable and are preferable for imaging applications.’

Additionally, we found and corrected some misprints in the text and improved English.

Reviewer 4 Report

The manuscript addresses a novel synthetic procedure to prepare silicon nanoparticles, which could be a good alternative for optical contrast agent or surface for optical sensor. The authors should address some questions before being able to accept the manuscript to publish:

  1. The authors should include characterization techniques such as high resolution TEM or XRD to support their statement about the high crystallinity of Si nanostructures. Because Raman spectrocopy is an excellent technique to characterize chemical composition and an excellent complementary technique for the crystallinity; however, only with Raman is complicate to obtain the degree of crystallinity such as the Si nanoparticles are monocrystaline or polycrystalline, etc.  
  2. I was wondering if HF used in the fabrication is completely eliminated after processing. The authors should characterize it because the presence of adsorbed F ions on the surface could be problematic in contact with aqueous solution due to their toxicity.

Author Response

Response to Reviewer 4 Comments

We thank the reviewer for high evaluation of the importance of our manuscript.

We have improved the manuscript text in accordance with the comments.

Point 1: The authors should include characterization techniques such as high resolution TEM or XRD to support their statement about the high crystallinity of Si nanostructures. Because Raman spectroscopy is an excellent technique to characterize chemical composition and an excellent complementary technique for the crystallinity; however, only with Raman is complicate to obtain the degree of crystallinity such as the Si nanoparticles are monocrystaline or polycrystalline, etc.  

Response 1: We agree. Raman spectroscopy is just one of various techniques to study composition and crystallinity of matter that does not allow us to define exactly crystalline or polycrystalline structure of the examined nanoparticles. Nevertheless, obtained Raman spectra indicate absence or negligible content of both amorphous silicon and polymorphous silicon nanocrystals in the examined particles at applying this technique.

The choice of Raman spectroscopy as a basic method for analysis in this work was due to following reasons. Raman spectra of silicon allows quantitative analysis of crystalline/amorphous phases of SiNPs with accuracy that can be hardly possible with XRD, which give us mainly qualitative analysis. Moreover, our attempts to study the SiNPs by XRD were failed due to relatively small concentration of the SiNPs for analysis. Meanwhile, Raman spectroscopy in a microscope mode with a high-resolution objective gave us reliable spectra. On the other hand, HR TEM does not allow us to get information on many nanoparticles simultaneously and, therefore, it demands a lot of time for comprehensive study.

The main aim of our study was not obtaining complete information of crystallinity of the formed nanoparticles, but mere demonstration the fact that the formed SiNPs contain rather great amount of crystalline phase (nano- and polycrystals) and a very small or negligible fraction of amorphous silicon, that is why we can limit our techniques by only Raman spectroscopy.

To avoid misunderstanding in terminology we changed the text by the following way:

  • Abstract, lines 31–33 ‘The essential features of the fabricated silicon nanoparticles are sizes smaller than 100 nm and crystalline phase presence.’
  • Page 4, lines 135–139 ‘Composition of the SiNPs was analyzed by measuring Raman spectra with a Horiba Jobin Yvon HR 800 spectrometer with excitation by Ar laser at the wavelength of 488 nm of the particles deposited on a glass substrate. High sensitivity in measuring small quantities of the SiNPs was ensured by using a 100x Olympus objective for this purpose.’
  • Page 5, lines 194–198 ‘Raman analysis of the SiNPs revealed long-range crystalline order in the SiNPs. A typical Raman spectrum (Figure 3b) demonstrates the narrow line near 521 cm-1 corresponding to the crystalline silicon phase and absence of both an amorphous fraction which is usually manifested by the wide line at 480 cm-1 [40] and other polymorphous Si fractions [41]. Most likely, the SiNPs are silicon nanocrystals or have polycrystalline structure.’
  • Page 8, lines 251–254 ‘Prevalence of the crystalline phase in the fabricated particles ensures their high refractive index (~3.6) [45] and fluorescence emission [32] in the so-called biotissue diagnostic transparency window (700 – 1300 nm), which indicates their high potential as scattering and/or fluorescent contrast agents in optical bioimaging.’
  • Conclusions, page 13, lines 408–411 ‘At the same time, the reported SiNPs, most likely, possess crystalline or polycrystalline structure confirmed by the analysis of Raman spectra. The relatively small sizes together with crystalline like structure open new prospects…’

Point 2: I was wondering if HF used in the fabrication is completely eliminated after processing. The authors should characterize it because the presence of adsorbed F ions on the surface could be problematic in contact with aqueous solution due to their toxicity.

Response 2: To minimize presence of HF remainders after electrochemical or metal-assisted chemical etching all samples were rinsed with water. It is a well-known technological step. Additionally, we emphasised it on page 3, lines 99–100: ‘After, the formed PSi layers were rinsed with distilled water to remove HF electrolyte remainders.’ In any case, the concentrations of the particles produced of PSi and SiNWs are of order 0.5 mg/ml and much less than the typical toxicity threshold for such structures in biological applications. We added the corresponding comments on pages 3–4, lines 125–127: ‘Such concentrations provide a priori good biocompatibility: for example, porous silicon based SiNPs could penetrate into the cells without any cytotoxic effect up to the concentration of 100 mg/ml [37].’

Additionally, we found and corrected some misprints in the text and improved English.

Round 2

Reviewer 4 Report

Thank you for considering my comments. I find your responses more than suitable. Recommend publication.